# The Influence of Tillage and Cover Cropping on Soil Microbial Parameters and Spring Wheat Physiology

**Alicja Niewiadomska [1], Leszek Majchrzak [2,*], Klaudia Borowiak [3], Agnieszka Wolna-Maruwka [1], Zyta Waraczewska [1], Anna Budka [4] and Renata Gaj [5]**

[1] Department of General and Environmental Microbiology, Poznań University of Life Sciences, Szydłowska 50, 60-656 Poznań, Poland; alicja.niewiadomska@up.poznan.pl (A.N.); amaruwka@up.poznan.pl (A.W.-M.); zyta.waraczewska@wp.pl (Z.W.)

[2] Department of Agronomy, Poznań University of Life Sciences, Dojazd 11, 60-632 Poznań, Poland

[3] Department of Ecology and Environmental Protection, Poznań University of Life Sciences, Piątkowska 94C, 60-649 Poznań, Poland; klaudine@up.poznan.pl

[4] Department of Mathematical and Statistical Methods, Poznań University of Life Sciences, Wojska Polskiego 28, 60-637 Poznań, Poland; anna.budka@up.poznan.pl

[5] Department of Agricultural Chemistry and Environmental Biogeochemistry, Poznań University of Life Sciences, Wojska Polskiego 71F, 60-625 Poznań, Poland; renata.gaj@up.poznan.pl

* Correspondence: leszek.majchrzak@up.poznan.pl

**Abstract:** The soil tillage system and the distribution of stubble catch crops increase the content of organic carbon, thus increasing the biochemical activity of soil. The aim of the study was to assess the impact of leguminous cover crops and different tillage soil systems before spring wheat sowing on the count of soil microorganisms, biochemical activity, microbiological diversity and the physiological state of the plants in correlation with yield. The study compared and analysed the following systems: (1) conventional tillage (CT) to a depth of 22 cm, followed by spring wheat sowing using four simplified cultivation technologies called conservation tillage. The following simplified tillage systems were evaluated: (2) skimming before sowing the cover crop and spring wheat sowing after ploughing tillage (CT), (3) skimming before sowing of the cover crop (sowing wheat with no-till technology (NT)), (4) direct sowing of ground cover plants (NT) and spring wheat sowing after ploughing cultivation (CT) and (5) direct sowing of cover crop (NT) and sowing wheat directly into cover crop (NT). The results showed that applying the cover crop and soil tillage method before sowing wheat improved all tested parameters. The highest values of the analysed parameters were observed in the treatment with soil skimming before sowing of the cover plant, and then with sowing the wheat directly into the mulch. The activity of dehydrogenase was 90% higher, while the activity of phosphatase was 32% higher, in comparison to the control group. Both the activity of catalase and the biological index of fertility were 200% higher, in comparison to the control group. Metagenomic analysis showed that soil bacterial communities collected during treatment 'zero' and after different cultivations differed in the structure and percentage of individual taxa at the phylum level.

**Keywords:** enzymatic soil activity; biological index of fertility; photosynthesis activity; biodiversity

---

## 1. Introduction

Reduced tillage, cover crops and fertilisers are some of the practices recommended for conservation agriculture (CA) in order to diminish soil erosion and greenhouse gas emissions. These practices are also beneficial for the improvement of soil and water quality, as well as crop productivity [1,2]. In recent years, conservation agriculture has become a global trend in agronomy. It consists of the

reduction of external outlays in agriculture and the dependence of relying on self-regulating processes triggered by soil microorganisms. Soil biology is considered a significant key element in modern agriculture. In contrast to the physical and chemical properties of soil, which tend to change slowly, the biological properties of soil are more sensitive [3]. The various methods of cultivation and all agrotechnical treatments that increase the amount of organic matter in soil and stimulate its biological functions are the basis of sustainable agriculture [4,5].

The importance of plants being grown as stubble cover crops is increasing in agricultural practice due to manure deficiency and the negative balance of organic matters in soil. The biomass of cover crops, especially when introduced into soil in the form of mulch or by tillage, has a complex effect on the properties of soil, and therefore, all subsequent sown crops. Cover crops are perceived as an element which protects soil from erosion and regenerates the field, especially if there is a high share of cereals in a crop rotation [6]. Cover crops mobilise other soil nutrients, such as nitrogen. They reduce the emission of greenhouse gases, and thus, reduce the causes of global warming. Additionally, leguminous cover crops ensure the biological fixation of nitrogen, which significantly affects the nitrogen balance in the entire crop rotation system.

As cover crops have a positive influence on the environment, they are currently an important element of the environment-friendly agricultural policy of the European Union, as well as an important agrotechnical treatment. When applied in appropriate amounts and at the correct time, cover crops not only provide nutrients to plants but also catalyse biochemical changes in soil, thus improving its fertility. Soil microorganisms transform the organic substances that are introduced into soil. Their biochemical activity is also an indicator of the degradation and transformation of organic matter occurring in soil, as well as the quality of soil itself. As a result of their positive effects, soil microorganisms increase the photosynthetic activity of plants, which results in a better yield. Earlier studies have shown that cultivation systems significantly influence the physiological activity of crops, i.e., their photosynthetic activity, stomatal conductance and transpiration [7].

Simplified soil cultivation and the share of the stubble catch crop in monocultural wheat cultivation increase the content of organic carbon, as well as the retention of water in soil. In consequence, the biochemical activity of soil increases. It is thought that catalases and dehydrogenases appear in soil as an integral part of intact living microbial cells. They are a measure of the overall microbial activity of soils and are used to determine the biological indicator of its fertility (BIF). These enzymes belong to the class of oxidoreductases. According to Ahmed et al. [8] and Bielińska and Płóciniak–Mocek [9], they fulfil the most important functions in the environment. Hydrolases are another important group of soil enzymes that include phosphatases, which participate in the phosphorus cycle. Other enzymes from the hydrolase group include ureases and β-glucosidases. Phosphatases are considered the most important indicators when it comes to changes in the physical conditions of soil. The count of microorganisms whose enzymes play an integral part must be monitored during cultivation treatments.

The main goal of this study was to assess the influence of the tillage systems and cover crops on the count of soil microorganisms (total bacteria, actinobacteria, fungi, oligotrophic bacteria and copiotrophic bacteria); the biochemical activity of soil (dehydrogenases, phosphatases and catalases) and the physiological condition of plants in correlation with the yield.

## 2. Materials and Methods

A field experiment was conducted at the Brody Research Station, Poznań University of Life Science in Poland (52°25′ N, 16°18′ E) on soil classified as Albeluvisol, which developed on loamy sands overlying loamy materials (12% clay, 19% silt and 69% sand). The static field experiment was conducted in 2014 and 2015 as a randomised block design in four replications using 20 plots (each with an area of 45 m$^2$). At the beginning of the experiment, at a depth of 0–20 cm, the soil layer contained 1.61% organic matter (the content of organic carbon was measured using the Tiurin method; the total nitrogen content was measured using the Kjeldahl method). The soil pH was 6.2 (measured in 1 M of potassium chloride (KCl)). The content of phosphorus (P), potassium (K) and magnesium

(Mg) amounted to 140, 160 and 40 mg/kg$^{-1}$, respectively. Arabella spring wheat (*Triticum aestivum* L.) cultivar was sown at a density of 500 seeds per 1 m$^2$ in all tillage treatments. The cultivar selected for tests is more resistant to powdery mildew and brown rust than other varieties. As Arabella plants are of medium height, they are more resistant to lodging. They also have lower protein and gluten contents in their seeds.

The study compared and analysed the following systems: (1) conventional tillage (CT) to a depth of 22 cm (control), followed by spring wheat sowing using four simplified cultivation technologies called conservation tillage. The following simplified tillage systems were evaluated: (2) skimming before sowing of the cover crop and spring wheat sowing after ploughing tillage (CT), (3) skimming before sowing of the cover crop (sowing wheat with no-till technology (NT)), (4) direct sowing of ground cover plants (NT) and spring wheat sowing after ploughing cultivation (CT) and (5) direct sowing of cover crop (NT) and sowing wheat directly into cover crop (NT) (Table 1). Yellow lupine (*Lupinus luteus* L.) and field pea (*Pisum sativum* L.) were used as cover crops.

**Table 1.** Experimental treatments.

| Treatment | Soil Tillage System Under Spring Wheat | Dose of Glyphosate + Adjuvant AS 500 SL and Time of Application | Soil Tillage System Under Cover Crop (Yellow Lupine + Field Pea) |
|---|---|---|---|
| 1 | CT (spring ploughing) | - | - |
| 2 | CT (spring ploughing) | - | CT (skimming) |
| 3 | NT (no tillage) | 1.5 L/ha$^{-1}$ + 1.5 L/ha$^{-1}$ (before spring wheat sowing) | CT (skimming) |
| 4 | CT (spring ploughing) | 4.0 L/ha$^{-1}$ + 1.5 L/ha$^{-1}$ (before legume sowing) | NT (no tillage) |
| 5 | NT (no tillage) | 4.0 L/ha$^{-1}$ + 1.5 L/ha$^{-1}$ (before legume sowing) 1.5 L/ha$^{-1}$ + 1.5 L/ha$^{-1}$ (before spring wheat sowing) | NT (no tillage) |

1—conventional tillage (CT) before sowing of spring wheat without a cover crop, 2—conventional tillage (CT) skimming before sowing of the cover crop and sowing of spring wheat after conventional tillage (CT), 3—skimming before sowing of the cover crop and sowing of spring wheat directly (no-till (NT)) into mulch, 4—sowing of the cover crop directly (NT) into the stubble and then sowing of spring wheat after conventional tillage (CT) and 5—sowing of the cover crop directly (NT) and sowing of spring wheat directly into mulch (NT).

Each year, the spring wheat straw that was harvested in the previous year was removed from all the plots. Legumes (yellow lupine and field pea) were sown as cover crops at rates of 100 + 100 kg/ha$^{-1}$ in the conventional tillage (CT) (after skimming) and no-till systems (NT). Glyphosate (N –(phosphonomethyl)glycine 360g/L-1) herbicide and AS 500 SL adjuvant were applied to the experimental area before sowing (Table 1).

Spring wheat was sown in the following treatments: (1) conventional tillage system (CT) and (2) no-till (NT) system with leguminous cover crops. The CT treatment consisted of tillage with a 22 cm-deep three-furrow reversible plough sown in the third week of March. It also consisted of presowing tillage with a field cultivator and was then followed by harrowing to a depth of 8 cm (one week before sowing) to prepare the seed bed. The NT treatment involved sowing seeds directly into the stubble of the previous crop. The CT plots were drilled with a traditional seed drill (width—2.5 m, row distance—15 cm), whereas the NT plots were drilled with a double-disc drill (Great Plains, solid stand 10 equipped with a fluted coulter for residue cutting, a double-disc for seed placement and a single press-wheel (width—3.05 m, row distance—17.8 cm). The operating speed for ploughing and drilling was 1.5 m/s$^{-1}$. For other tillage treatments (cultivator and disc harrow) the operating speed was 1.8 m/s$^{-1}$. The spring wheat sowing dates were dependent on the soil water conditions—late March. The seeds were sown at a depth of 3–4 cm in all tillage systems.

The principles of good agricultural practice were followed in all chemical treatments (fertilisation and herbicide protection) applied to spring wheat. The treatments were not a research factor.

The same fertilisation treatment was applied to all tillage systems (90 kg nitrogen (N)/ha$^{-1}$, 24 kg P/ha$^{-1}$ and 24 kg K/ha$^{-1}$) in both years of the experiment.

A preplant and postemergence herbicide programme was applied to all tillage systems. Glyphosate herbicide (1.5 L/ha$^{-1}$) and AS 500 SL adjuvant (1.5 L/ha$^{-1}$) were applied to all no-till plots to control perennial weeds and volunteers before planting. During the growing season, after the emergence of plants, Lintur 70% WG (Syngenta Crop Protection AG, Basel, Switzerland) herbicide (dicamba 65.9% + triasulfuron 4.1%) + Chwastox Extra 300 soluble liquid (SL) (CIECH Sarzyna, Nowa Sarzyna, Poland) herbicide (2-methyl-4-chlorophenoxyacetic acid (MCPA) 300 g/L$^{-1}$) herbicide were applied at the rate of 150 g/ha$^{-1}$ + 1.0 L/ha$^{-1}$ for weed control. The seeds were treated with Raxil Extra 060 flowable concentrate (FS) (Bayer CropScience AG, Germany) fungicide (0.06 L per 100 kg seeds) containing thiuram and tebuconazole. For disease control, Falcon 460 emulsifiable concentrate (EC) (Bayer CropScience AG, Leverkusen, Germany) fungicide (spiroksamine 250 g/L$^{-1}$ + tebuconazole 167 g/L$^{-1}$ + triadimenol 43 g/L$^{-1}$) was applied at a rate of 0.6 L/ha$^{-1}$ in all plots at the biologische, bundesanstalt, bundessortenamt and chemical (BBCH) 32 growth stage. Fury 100 (oil-in-water emulsion (EW) (FMC Chemical, Brussels, Belgium) insecticide (zeta–cypermetryne 100 g/L$^{-1}$) was applied with a dose of 0.1 L/ha$^{-1}$ at the BBCH 61–65 growth stages.

## 2.1. Microbiological Analyses

Soil samples for biochemical and microbiological analyses were collected from the arable layer (0–20 cm) during four terms corresponding to the successive phases of wheat development. The terms were as follows: 1st term—emergence (BBCH 16–17), 2nd term—flowering (BBCH 61–65) in late June and early July, 3rd term—technological ripeness of seeds (BBCH 87–89) and 4th term—post-harvest. Soil samples were randomly collected from five locations between rows of each experimental plot. There were four replications for each of the five treatments of the experiment. Thus, there were 20 one-kilogram samples of soil collected during each term of analyses. The soil was sieved before microbiological and biochemical analyses.

### 2.1.1. Counts of Microorganisms

The count of microorganisms in soil samples collected from under the plants at a depth of 0–20 cm was measured by means of series dilution on appropriate agars (with five replicates). The average count of the following colonies of microorganisms per dry mass of soil was measured:

- total bacterial count—on ready-made Merck standard agar after 5 days of incubation at 25 °C,
- fungi—on Martin agar [10] after 5 days of incubation at 24 °C,
- copiotrophs—on nutrient broth (NB) agar [11] after 5 days of incubation at 25 °C,
- oligotrophs—on diluted nutrient broth (DNB) agar [11] after 5 days of incubation at 25 °C and
- actinobacteria—on Pochon agar after 5 days of incubation at 25 °C [12].

### 2.1.2. Analyses of Enzymatic Activity

The enzymatic activity of soil enzymes in the experimental treatments was measured using the following methods:

- dehydrogenases (EC 1.1.1.)—colorimetry with 1% TTC (triphenyltetrazolium chloride) as a substrate after 24-hour incubation at 30 °C at a wavelength of 485 nm, expressed as μmol triphenylformazan (TPF) kg$^{-1}$/24 h$^{-1}$ [13];
- acid and alkaline phosphatase (EC 3.1.3.2)—spectrophotometry with sodium p-nitrophenyl phosphate as a substrate after 1-hour incubation at 37 °C at a wavelength of 400 nm, expressed as μmol para-nitrophenol (PNP) g$^{-1}$dm/h$^{-1}$ (Novospac spectrophotometer) [14] and
- catalase (EC 1.11.1.6)—permanganometry with 0.3% hydrogen peroxide (H$_2$O$_2$) as a substrate after 20 minutes incubation at room temperature (about 20 °C), titrated with 0.02 M potassium permanganate (KMnO$_4$) to light pink colour and expressed as mmol H$_2$O$_2$ g$^{-1}$dm/min$^{-1}$ [15].

### 2.1.3. Biological Index of Fertility

Biological index of fertility (BIF) was calculated on the basis of the dehydrogenase activity (DHA) and catalase activity (CAT), according to the following formula: (DHA + kCAT)/2, where k is the proportionality factor, which amounts to 0.01 [16].

### 2.2. Genetic Analyses

Innovative metagenomic tools were used in order to precisely determine the influence of simplified cover crop cultivation methods, as well as the wheat sowing method, on the biodiversity of soil bacteria. The metagenomic tools provided detailed information on the domination of specific types of bacteria in individual cultivation treatments of the field experiment. Soil samples were collected before the experiment (sample 'zero') and from individual experimental treatments during the harvest (Samples 1–5, Table 1). Afterwards, the new-generation sequencing technology Ion Torrent personal genome machine (PGM) (Life Technologies) was applied. The 16S rRNA encoding gene was analysed metagenomically on the basis of the V3–V4 hypervariable region. Specific primer sequences 341F and 785R were used to amplify the selected region and prepare the library. The polymerase chain reaction was carried out with a Q5 Hot Start High-Fidelity DNA Polymerase (NEBNext) following instructions recommended by the manufacturer.

An MiSeq sequencer was used for paired-end (PE) sequencing (2 × 250 nucleotide (nt)) using the MiSeq Reagent Kit v2, as recommended by the manufacturer (Illumina). Detailed information can be found on the websites of the reagent manufacturers.

The MiSeq apparatus with the MiSeq reporter software (MSR) v2.6, 16S Metagenomics protocol, was used for automatic data analysis, which consisted of three stages:

1. automatic demultiplexing of the samples,
2. generating FASTQ files containing raw reads and
3. classification of paired-end reads into individual taxonomic categories.

### 2.3. Physiological State of Plants

The handheld photosynthesis system Ci 340aa (CID BIOSCIENCE Inc., Camas, WA, USA) was used to measure the net photosynthetic rate ($P_N$), stomatal conductance ($g_s$), transpiration rate ($E$) and intercellular carbon dioxide ($CO_2$) ($C_i$) concentration. Measurements in the leaf chamber were collected at constant values. These values were as follows: $CO_2$ inflow concentration—390 μmol ($CO_2$) mol$^{-1}$, photosynthetic photon flux density (PPFD)—1000 μmol (photon) m$^{-2}$/s$^{-1}$, chamber temperature—25 °C and relative humidity—40 ± 3%. The measurements were collected once a year using five plants selected from each experimental treatment.

### 2.4. Statistical Analysis

The R and STATISTICA 12.0 (StatSoft Inc., Poland, Krakow) software package was used for all statistical analyses. The effect of the experimental factors (tillage system and cover crop) on the count of soil microorganisms and the enzymatic activity of soil was tested using two-way ANOVA. The symbol $y_{ij}$ expressed the estimated value of the variables (the count of soil microorganisms and the enzymatic activity of soil) from the analysed *i*-observation terms ($i = 1 \ldots, 4$) using different tillage systems and cover crops ($j = 1 \ldots, 5$) expressed by *j*.

The model for the two-way ANOVA, which included interactions of the factors, was as follows for constant factors A and B:

$$y_{ij} = \mu + \alpha_i + \beta_j + (\alpha\beta)_{ij} + e_{ij} \tag{1}$$

where $\mu$ is grand mean, $\alpha_i$ is the effect of $i^{th}$ analysed terms, $\beta_j$ is $j^{th}$ different tillage systems and cover crops and $(\alpha\beta)ij$ is A and B interaction effect at $\alpha_i \beta_j$.

Physiological plant parameters were tested once a year for the experiment. The mixed-effects model in two-way ANOVA without interactions of the factors was used for random factor 'year' and constant factor 'treatment'. The interaction of the years with the treatments was not significant; therefore, it was omitted in further analyses. The Tukey's test (multiple comparison procedure) was used for comparing mean physiology parameters under different tillage systems and cover crops at a significance level of $\alpha = 0.05$

Principal component analysis (PCA) was used to analyse the microbial activity of soil in five different tillage systems. The tillage systems used in the analysis were as follows:

1.  conventional tillage (CT) before sowing of spring wheat without a cover crop,
2.  CT skimming before sowing of the cover crop and sowing of spring wheat after CT,
3.  skimming before sowing of the cover crop and sowing of spring wheat directly (no-till (NT)) into mulch,
4.  sowing of the cover crop directly (NT) into the stubble and then sowing of spring wheat after CT and
5.  NT and sowing of spring wheat directly into mulch (NT).

The heat map and cluster analyses were based on the mean values noted during the two years of the experiment. Similarities between the microbial soil activity, gas exchange parameters and the experimental treatments and yield were analysed. A cluster analysis was conducted to group similar treatments according to all the considered parameters. Euclidean distance measurements and Ward's hierarchical clustering were used to determine the dendrogram.

## 3. Results and Discussion

### 3.1. Weather Data

During the study, the average air temperature ranged from 8.6 to 10.0 °C. The values were close to the average temperatures over a period of many years. Total rainfall, especially the distribution of rainfall, is also an important factor for optimal growth conditions and yield. The values of the Sielianinov index reflected the diversity of the weather conditions in the study years (Figure 1). This index is a measure of rainfall effectiveness. It shows dry periods and optimal periods for plant growth [17]. In the diagram, the Sielianinov index indicates humidity during the growth of wheat in individual years of the experiment. In 2014, especially in May, humidity was more favourable for the growth of wheat than in 2015 (K = 0.9).

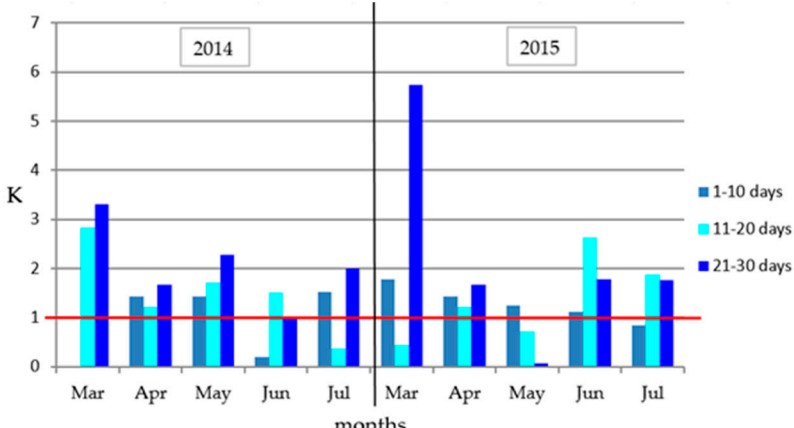

**Figure 1.** Value of Sielianinov's hydrothermal coefficient in study years. Interpretation: K > 1.5 indicates excessive moisture for all plants, K = 1.0–1.5 indicates sufficient moisture, K = 0.5–1.0 indicates insufficient moisture and K < 0.5 indicates moisture less than the requirement for most of plants (drought).

Table 2 denotes the temperature and rainfall during the experiment. The data were correlated with the microbiological parameters of soil, such as the biochemical activity and the count of microorganisms in various treatments of the experiment.

**Table 2.** The weather conditions during the experiment. BBCH = biologische, bundesanstalt, bundessortenamt and chemical.

| Month | Days | Term of Analyses | Average Temperature (°C) | | Average Rainfall (mm) | |
|---|---|---|---|---|---|---|
| | | | 2014 | 2015 | 2014 | 2015 |
| March | 1–10 | | 4.3 | 4.7 | 0.0 | 8.4 |
| | 11–20 | | 7.7 | 5.0 | 21.8 | 2.2 |
| | 21–31 | | 7.9 | 5.8 | 26.0 | 33.3 |
| | Average/Total | | 6.6 | 5.2 | 47.8 | 43.9 |
| | Long-term | | 2.9 | | 39.2 | |
| April | 1–10 | | 8.8 | 4.8 | 12.5 | 25.0 |
| | 11–20 | | 8.7 | 9.4 | 10.5 | 0.3 |
| | 21–30 | | 13.9 | 11.5 | 23.3 | 6.7 |
| | Average/Total | | 10.5 | 8.6 | 46.3 | 32.0 |
| | Long-term | | 8.1 | | 37.4 | |
| May | 1–10 | | 10.1 | 12.8 | 14.5 | 15.9 |
| | 11–20 | 1st term (BBCH 16–17) | 12.6 | 12.6 | 21.4 | 8.9 |
| | 21–31 | | 16.5 | 13.6 | 37.6 | 0.8 |
| | Average/Total | | 13.1 | 13.0 | 73.5 | 25.6 |
| | Long-term | | 13.2 | | 57.4 | |
| June | 1–10 | | 17.5 | 16.3 | 3.3 | 18.3 |
| | 11–20 | | 16.0 | 15.4 | 24.0 | 40.3 |
| | 21–30 | 2nd term (BBCH 61–65) | 14.8 | 15.0 | 14.7 | 26.7 |
| | Average/Total | | 16.1 | 15.5 | 42.0 | 85.3 |
| | Long-term | | 16.6 | | 63.8 | |
| July | 1–10 | | 20.5 | 20.3 | 31.0 | 17.0 |
| | 11–20 | 3rd term (BBCH 87–89) | 21.7 | 19.1 | 7.8 | 35.8 |
| | 21–31 | 4th term after the harvest | 22.2 | 18.2 | 44.3 | 32.1 |
| | Average/Total | | 21.5 | 19.2 | 83.1 | 84.9 |
| | Long-term | | 18.3 | | 81.3 | |

In the first year of the experiment, there were optimal humidity conditions for the growth of wheat. These conditions were optimal from the moment of sowing until the end of May. However, the rainfall in early June was only 3.3 mm, whereas in mid and late-June, it was 24 mm and 14.7 mm, respectively. In total, there was a rainfall deficit of 21.8 mm, in comparison with the average long-term rainfall for this period. In 2015, the humidity conditions were favourable from the wheat emergence until the BBCH phase 16–17. There was a drought, especially in mid and late-May. In comparison with the long-term average for this period, the difference in rainfall amounted to 31.8 mm. By contrast, in June and July, the rainfall was higher than the long-term average. In both years, temperatures were higher during the wheat-growing season than the long-term average in March, April and July. In 2014, the soil was more humid during the first and fourth terms of sample collection (spring wheat BBCH 16–17 and post-harvest). In 2015, there was higher rainfall during the second and third terms of sample collection (wheat BBCH 61–65 and 87–89).

### 3.2. Soil Organic Matter Content

According to other authors, stubble crops and simplified cultivation affect the content of organic substances in soil. In our experiment, the direct sowing technology was applied for the catch

crop (yellow lupine + field pea) and then for spring wheat. In consequence, the organic carbon content was higher than before the sowing of wheat (Table 3). The same situation was observed in the treatments where the stubble catch crop was sown directly, followed by ploughing before wheat sowing. We observed an increase in carbon content after the direct sowing of wheat. This was consistent with the findings of a study conducted by Thomas et al. [18] and Lopez–Fando and Pardo [19]. They concluded that the higher content of organic carbon in soil in the simplified cultivation system resulted from slower mineralisation of organic matter and lower aeration of soil.

**Table 3.** Soil chemical properties (means for years 2014–2015).

| Specification | Investigation Term | Treatments | | | | |
|---|---|---|---|---|---|---|
| | | 1 * | 2 | 3 | 4 | 5 |
| C organic (g/kg$^{-1}$ soil) | A ** | 10.8 | 12.9 | 10.1 | 10.9 | 10.4 |
| | B ** | 10.7 | 11.8 | 11.0 | 12.4 | 11.9 |
| N total (g/kg$^{-1}$ soil) | A | 0.7 | 1.1 | 1.0 | 1.0 | 1.2 |
| | B | 0.9 | 1.2 | 1.0 | 1.3 | 0.9 |
| C:N | A | 12.6 | 14.4 | 11.9 | 11.8 | 10.2 |
| | B | 10.5 | 13.2 | 12.3 | 10.0 | 14.3 |

* see Materials and Methods, experimental design; ** A—before the experiment; ** B—during the experiment; 1—conventional tillage (CT) before sowing spring wheat without a cover crop; 2—conventional tillage (CT) skimming before sowing of the cover crop and sowing of spring wheat after conventional tillage (CT); 3—skimming before sowing of the cover crop and sowing of spring wheat directly (no-till (NT)) into mulch; 4—sowing the cover crop directly (NT) into the stubble and then sowing spring wheat after conventional tillage (CT) and 5—sowing the cover crop directly (NT) and sowing spring wheat directly into mulch (NT). C = carbon and N = nitrogen.

The biggest changes in the total nitrogen content after wheat harvest were found in the fourth treatment (sowing the cover crop directly (NT) into the stubble and then sowing spring wheat after conventional tillage) and the fifth treatment (sowing the cover crop directly (NT) and sowing spring wheat directly into mulch (NT). The total nitrogen content increased by 30% in the fourth treatment and decreased by 25% in the fifth treatment, when compared to the values prior to the study.

The carbon to nitrogen ratio (C:N) was also measured. The value of the ratio varied according to the treatment and measurement period. After the two-year experiment, the C:N ratio increased only in Treatments 3 (skimming before sowing the cover crop and sowing spring wheat directly (no-till (NT)) into mulch) and 5 (sowing the cover crop directly (NT) and sowing spring wheat directly into mulch (NT)). However, it tended to decrease in the other treatments. The C:N ratio was the lowest in the treatment with conventional cultivation, which points to intensified mineralisation processes.

### 3.3. The Count of Microorganisms

The type of soil cultivation had a significant impact on the count of microorganisms, the qualitative selection of entire groups of microorganisms and the process of photosynthesis, as well as crop yield.

Incorrect agrotechnical treatments may gravely disturb the functioning of agroecosystems. The research showed that sowing the cover crop and the cultivation of spring wheat influenced the count of microorganisms, which are of key significance to the proper functioning of the soil ecosystem. Their main task is to control the reactions that are necessary to maintain the correct structure and fertility of soil.

Our results showed that the use of cover crops, as well as the method of its cultivation, had an impact on the count of individual groups of soil microorganisms, which included total bacteria, oligotrophs, copiotrophs, fungi and actinobacteria. Other factors that had an impact included the method of soil cultivation before sowing of wheat after the cover crop and the terms of analyses.

The two-way analysis of variance showed that applying the cover crop and the soil cultivation method before sowing of wheat had a significant influence ($\alpha = 0.05$) on the diversity of selected groups of soil microorganisms (Table 4).

**Table 4.** The test F statistics and the significance level of two-way analysis of treatments for the number of selected groups of soil microorganisms. The traits analysed were affected by two factors, i.e., cultivation system and the terms of the test.

| Parameter | Term | Treatment | Interaction |
|---|---|---|---|
| Total count of bacteria | 87.78 * | 120.67 * | 99.67 * |
| *Actinobacteria* | 58.27 * | 21.60 * | 17.58 * |
| *Fungi* | 54.53 * | 74.44 * | 85.66 * |
| *Oligotrophic bacteria* | 45.96 * | 23.6 * | 35.58 * |
| *Copiotrophic bacteria* | 89.89 * | 101.92 * | 111.99 * |

F test statistics and significance levels of two-way analysis of variance for the number of microorganisms associated with the cultivation system and terms of research fixed factors. * is $p = 0.05$.

Additionally, the terms during which the study was conducted significantly influenced all groups of microorganisms analysed. The method of sowing the stubble cover crop and the method of soil cultivation before sowing wheat significantly impacted the count of the soil microorganisms studied.

Since the impact of wheat cultivation on the count of soil microorganisms was similar in both years of the study, the results were presented as mean values referring to the study years (Table 5).

**Table 5.** The count of microorganisms.

| Experimental Treatment | Term of Analysis | | | |
|---|---|---|---|---|
| | 1st Term Emergence (BBCH 16–17) | 2nd Term Flowering (BBCH 61–65) | 3rd Term Technological Ripeness of Seeds (BBCH 87–89) | 4th Term After Harvest |
| *Total count of heterotrophic bacteria* (cfu g$^{-1}$/dm soil $10^5$) | | | | |
| 1 | 4.7 ± 3.8 | 6.3 ± 1.9 | 6.4 ± 2.4 | 14.5 ± 4.6 |
| 2 | 30.8 ± 12.1 | 20.3 ± 20.8 | 22.5 ± 3.8 | 45.5 ± 5.1 |
| 3 | 30.3 ± 7.1 | 55.4 ± 12.3 | 37.9 ± 7.6 | 26.5 ± 5.0 |
| 4 | 22.8 ± 8.4 | 15.0 ± 2.3 | 12.8 ± 4.3 | 40.4 ± 5.2 |
| 5 | 162.3 ± 52.8 | 15.9 ± 4.1 | 14.6 ± 4.5 | 37.9 ± 5.2 |
| *Actinobacteria* (cfu g$^{-1}$/dm soil $10^4$) | | | | |
| 1 | 232.9 ± 58.0 | 243.1 ± 47.0 | 193.8 ± 30.6 | 175.3 ± 3.2 |
| 2 | 204.8 ± 79.8 | 231.4 ± 38.5 | 117.2 ± 27.5 | 165.7 ± 5.0 |
| 3 | 126.2 ± 21.8 | 247.9 ± 33.0 | 133.6 ± 27.8 | 136.0 ± 14.6 |
| 4 | 131.7 ± 45.9 | 149.2 ± 24.9 | 43.4 ± 13.7 | 129.7 ± 5.9 |
| 5 | 202.5 ± 31.0 | 125.2 ± 21.4 | 55.7 ± 18.1 | 130.6 ± 4.6 |
| *Fungi* (cfu g$^{-1}$/dm soil $10^4$) | | | | |
| 1 | 0.3 ± 0.0 | 1.4 ± 0.6 | 1.4 ± 1.3 | 1.1 ± 0.7 |
| 2 | 1.5 ± 2.1 | 2.6 ± 0.9 | 2.3 ± 1.7 | 3.4 ± 1.0 |
| 3 | 0.6 ± 0.6 | 3.9 ± 0.1 | 4.1 ± 3.8 | 3.0 ± 1.2 |
| 4 | 3.2 ± 4.1 | 2.4 ± 0.2 | 3.8 ± 1.9 | 4.2 ± 1.2 |
| 5 | 33.2 ± 16.4 | 2.9 ± 0.3 | 2.0 ± 1.3 | 2.6 ± 2.1 |
| *Oligotrophic bacteria* (cfu g$^{-1}$/dm soil $10^5$) | | | | |
| 1 | 35.8 ± 9.7 | 49.7 ± 9.2 | 33.0 ± 3.9 | 62.4 ± 2.5 |
| 2 | 37.1 ± 13.9 | 41.1 ± 5.5 | 22.3 ± 5.5 | 33.4 ± 3.1 |
| 3 | 38.6 ± 10.7 | 32.1 ± 3.6 | 33.3 ± 8.8 | 49.9 ± 5.0 |
| 4 | 56.9 ± 18.0 | 23.7 ± 5.3 | 31.6 ± 9.6 | 54.5 ± 5.6 |
| 5 | 150.3 ± 27.4 | 23.5 ± 9.1 | 32.6 ± 5.2 | 60.0 ± 2.5 |

**Table 5.** *Cont.*

| Experimental Treatment | Term of Analysis | | | |
| --- | --- | --- | --- | --- |
| | 1st Term Emergence (BBCH 16–17) | 2nd Term Flowering (BBCH 61–65) | 3rd Term Technological Ripeness of Seeds (BBCH 87–89) | 4th Term After Harvest |
| *Copiotrophic bacteria* (cfu g$^{-1}$/dm soil 10$^5$) | | | | |
| 1 | 14.1 ± 8.3 | 19.9 ± 4.6 | 12.2 ± 5.0 | 10.9 ± 5.2 |
| 2 | 14.6 ± 7.4 | 19.3 ± 5.4 | 19.1 ± 2.1 | 11.7 ± 5.3 |
| 3 | 20.2 ± 6.3 | 27.5 ± 7.2 | 21.9 ± 3.7 | 15.8 ± 5.5 |
| 4 | 10.1 ± 6.1 | 25.1 ± 6.5 | 13.3 ± 4.8 | 25.9 ± 5.9 |
| 5 | 74.6 ± 11.2 | 26.2 ± 5.2 | 15.2 ± 4.3 | 20.6 ± 5.5 |

1—conventional tillage (CT) before sowing spring wheat without a cover crop, 2—conventional tillage (CT) skimming before sowing the cover crop and sowing spring wheat after conventional tillage (CT), 3—skimming before sowing the cover crop and sowing spring wheat directly (no-till (NT)) into mulch, 4—sowing the cover crop directly (NT) into the stubble and then sowing spring wheat after conventional tillage (CT) and 5—sowing the cover crop directly (NT) and sowing spring wheat directly into mulch (NT). cfu—colony-forming unit.

During the cultivation of wheat, the total count of heterotrophic bacteria was higher in all the experimental treatments. A difference was observed between the total count in the treatment where the plants were grown conventionally without the cover crop sown before and the control group. The highest total count of heterotrophic bacteria was observed at the plants' emergence phase in the treatment where the cover crop and wheat were sown directly (NT) into stubble mulch (Treatment 5). During the second (flowering) and third terms (full technological ripeness of the plants), the highest count of heterotrophic bacteria was noted in the soil that was skimmed before the cover crop was sown and where spring wheat was grown after tillage (Treatment 3). The count of these microorganisms amounted to 55.4 colony-forming unit (cfu) g$^{-1}$/dm of soil during the second term and 37.9 cfu g$^{-1}$/dm of soil during the third term (Table 5).

Similarly, fungi and copiotrophic bacteria reacted in an analogous manner to the method of cover crop cultivation and the method of sowing wheat after the cover crop. However, there were no similar dependencies found for oligotrophic bacteria and actinobacteria (Table 5).

During the first term of analyses, where the cover crop was sown directly (NT) and wheat was also sown directly (NT) into the stubble mulch (Treatment 5), the count of fungi was 99% greater, whereas the count of copiotrophic bacteria was 81% greater when compared to the control treatment. During the subsequent terms of analyses, i.e., during flowering and the technological maturity of seeds, these groups of microorganisms were most stimulated in the third treatment (soil skimming before sowing the cover crop and direct sowing of spring wheat). Under these conditions, the counts of fungi and copiotrophic bacteria reached their highest values, as compared with the control treatment where conventional cultivation was applied and no cover crop was sown.

During the last term of microbiological analyses, i.e., after the harvest, the method of cover crop cultivation and wheat sowing stimulated only some groups of microorganisms (total bacterial count, copiotrophs and fungi). In each experimental treatment of simplified cultivation, the counts of these microorganisms were greater than in the conventional tillage (control) treatments (Table 5). However, the count of actinobacteria and the physiological group of oligotrophic bacteria in the conventional tillage treatments were greater than in the simplified technology treatments.

During the cultivation of wheat, the count of fungi was higher in all the experimental treatments. Results obtained from other studies showed that fungi are important members of soil microbial communities in row crops. They provide essential ecosystem benefits, such as nutrient cycling, organic matter decomposition and increased soil structure. However, fungi are also more sensitive to physical disturbances than other microorganisms. The implementation of conservation management practices, such as no-till and cover cropping, shape the structure and function of soil fungal communities. No-till eliminates or greatly reduces the physical disturbances that redistributes organisms and

nutrients in the soil profiles and disrupts fungal hyphal networks, while cover crops provide additional types and greater abundance of organic carbon sources [20]. However, in our experiment, there was not much difference in the density of cultivable fungi between the till and no-till treatments. This could be explained by the fact that the fungi identified on Martin's medium were mainly saprophytes. As research shows, the use of cover crops increases species diversity, while no-till shifts the symbiotroph to saprotroph ratio in favour of symbiotrophs. These management-induced shifts in the composition of the fungal community could lead to greater resilience of the ecosystem and provide crops with greater access to limiting resources. Substrates with higher contents of nutrients are used to estimate the count of symbiotrophs, including mycorrhizal fungi and fungi of the *Trichoderma* genus.

The results of studies presented in reference publications show that the causes of the subsequent effects of the cover crop or forecrop on the cultivated species of cereals and bioactivity of soils are diversified, complicated and, thus far, they have not been thoroughly explained [21,22].

Arcand et al. [23] observed changes in the microbial community in the soils where organic residues were supplied. The authors pointed to the dominance of fungi and heterotrophic bacteria, which actively degraded complex compounds such as cellulose, lignin and chitin. These compounds constitute a relatively large part of cover crop stubble mulch. The presence of nutrients in soil, which results from handling the cover crops, affects the rate of metabolism, which is correlated with larger populations of selected physiological groups of microorganisms in soil [24]. Moreover, as published research results indicate, during the growth of plants cultivated as cover crops before cereals, substances and organic compounds with subsequent phytotoxic effects are released into the soil. It is also known that during the decomposition and mineralisation of plant residues (roots, stems, straw and chaff), various organic compounds, including phytotoxic ones, are formed and accumulated in soil. These compounds may stimulate or inhibit the development of soil microbiota.

### 3.4. Influence of Various Cultivation Methods on Taxonomic Distribution of Soil Bacteria

Along with the analyses of bacterial cultures, the soil microbiome was analysed on the basis of the 16S rRNA gene to determine metapopulation changes. The analysis revealed that when different spring wheat cultivation systems and different methods of forecrop sowing were applied, over 99% of the microorganisms in each soil sample collected from individual experimental treatments belonged to the *Bacteria* kingdom, whereas 0.01–0.02% belonged to the *Archaea* kingdom. The analysis also revealed the presence of microorganisms which have not been classified yet (0.02–0.03%).

The research results show that the communities of soil bacteria in the samples collected at term 'zero' and in the ones collected after the harvest, where the treatment-specific methods of cover crop cultivation and wheat sowing had been applied, differed in the structure and percentage share of individual taxa, categorised as the phylum of microorganisms (Figure 2).

The metagenomic analysis of the 16S rRNA soil metabiome showed the dominance of three types of bacteria: *Actinobacteria*, *Proteobacteria* and *Firmicutes*, in all the experimental treatments (Figure 2). This finding is in agreement with reference publications concerning the cultivation of field soils [25,26].

On average, there were about 40% of *Actinobacteria* sequences (29.69–62% of all bacterial sequences found in the investigations) both in the samples of conventionally cultivated soil and in the ones where simplified wheat cultivation technologies were applied (Figure 2). The highest percentage of *Actinobacteria* was found in sample 'zero', which was collected before the experiment. During the harvest, the highest percentage of *Actinobacteria* was found in the first treatment, where conventional tillage was used. The genetic analyses revealed the presence of bacterial subdominants in the soil, i.e., *Proteobacteria* (18.10–35.79%) and *Firmicutes* (8.71–13.78%). On the other hand, the occurrences of *Chloroflexi*, *Cyanobacteria*, *Gemmatimonadetes*, *Planctomycetes*, *Spirochaeta*, *Synergistetes* and *Verrucomicrobia* sequences were less frequent. The share of each of these types was under 1%. The *Actinobacteria* type was dominant in sample 'zero' (collected before soil cultivation), where it amounted to 62.23%. The shares of *Proteobacteria* and *Firmicutes* amounted to 19.04% and 8.73%, respectively (Figure 2).

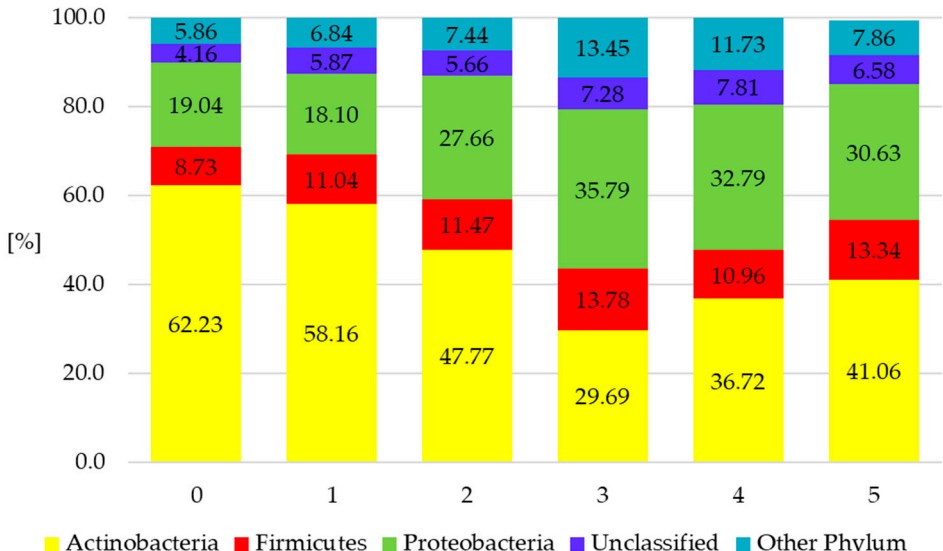

**Figure 2.** The composition of the structure of bacterial communities in the soil (phylum level classification interval ≥1% of sequences). 0—sample 'zero' before sowing, 1—conventional tillage (CT) before sowing spring wheat without a cover crop, 2—conventional tillage (CT) skimming before sowing the cover crop and sowing spring wheat after conventional tillage (CT), 3—skimming before sowing the cover crop and sowing spring wheat directly (no-till (NT)) into mulch, 4—sowing the cover crop directly (NT) into the stubble and then sowing spring wheat after conventional tillage (CT) and 5—sowing the cover crop directly (NT) and sowing spring wheat directly into mulch (NT).

The content of *Actinobacteria* was, on average, 4–32% lower in the soils where specific methods of simplified wheat cultivation and cover crop sowing were applied. However, the content of the *Proteobacteria* type in these soils increased to about 16% on average, as compared with the uncultivated soil (sample 'zero'). Moreover, the percentage share of the *Firmicutes* bacterial type, which includes probiotic lactic acid bacteria and *Arthrobacter* bacteria, increased in the experimental treatments where soil cultivation before sowing wheat was limited and where the cover crop was applied.

The results of our experiment are in line with the results of studies presented in reference publications. Wang et al. [27] observed that the counts of *Firmicutes* and *Proteobacteria* in simplified tillage technologies were larger than in conventional tillage cultivation. Navarro–Noya et al. [26] observed similar dependencies. They found that soil conservation treatments resulted in better water and air conditions and a higher content of nutrients than conventional cultivation. The treatments provided adequate conditions for the development of the *Proteobacteria* and *Firmicutes* bacterial types.

Considering the entire composition of bacterial communities, our study revealed a significant correlation between the types of bacteria detected with genetic methods and those identified by means of culturing (Figure 3).

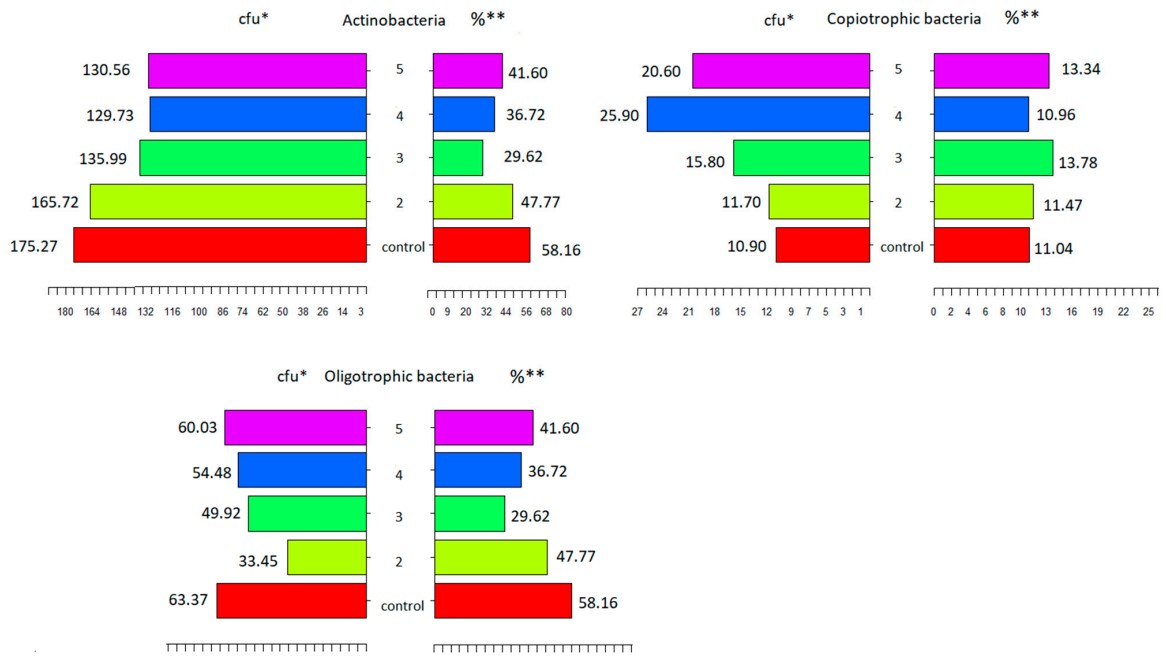

**Figure 3.** The relation between the count of microorganisms measured with different methods. Abbreviations: cfu *—colony-forming unit, the count of microorganisms measured on selective culture media; % **—the percentage of microorganisms measured by metagenomic analysis; control—conventional tillage (CT) before sowing spring wheat without a cover crop; 2—conventional tillage (CT) skimming before sowing the cover crop and sowing spring wheat after conventional tillage (CT); 3—skimming before sowing the cover crop and sowing spring wheat directly (no-till (NT)) into mulch; 4—sowing the cover crop directly (NT) into the stubble and then sowing spring wheat after conventional tillage (CT) and 5—sowing the cover crop directly (NT) and sowing spring wheat directly into mulch (NT).

Fierer et al. [28] proposed the concept of copiotrophic and oligotrophic bacteria, in which the *Firmicutes* and *Proteobacteria* types were described as rapidly growing copiotrophs that developed in environments with high availability of carbon and which were characteristic of conservation crops. Rodrigues et al. [29] indicated similar dependencies in the development of *Firmicutes* and *Proteobacteria*, which exhibited a positive reaction to the supply of carbon in the soil. On the other hand, the *Actinobacteria* and *Acidobacteria* types are considered oligotrophs [30], and they are characteristic of the soils where conventional tillage is used. The statistical analysis (Figure 3) confirmed the abovementioned dependencies observed in our experiment, where *Firmicutes* and *Proteobacteria*, categorised as copiotrophs, were the dominant bacterial types in the simplified cultivation treatments. Their percentage shares corresponded to the colony-forming units of copiotrophs. The percentage of *Actinobacteria*, the dominant type in the conventional tillage treatment, also interfered with the colony-forming units in the culturing method. The data in Figure 3 indicates the same dependencies between the communities of soil bacteria, regardless of the methodology used for assessment of the microbial activity.

*3.5. Biochemical Analyses of Soil*

The two-way analysis of variance showed that the cover crop and the soil cultivation method applied before sowing wheat had highly significant influence ($\alpha = 0.05$) on the biological index of fertility (BIF) and the activity of selected soil enzymes, except acid phosphatase (Table 6).

**Table 6.** The test F statistics and the significance level of two-way analysis of treatments for the enzymatic activity. The traits under analysis were affected by two factors, i.e., cultivation system and the term of the test.

| Parameter | Term | Treatment | Interaction |
|---|---|---|---|
| Dehydrogenase activity (DHA) | 45.3 * | 11.2 * | 4.9 * |
| Acid phosphatase activity (PAC) | 25.5 * | 1.4 ns | 3.9 * |
| Alkaline phosphatase activity (PAL) | 23.3 * | 1.4 * | 9.96 * |
| Catalase activity (CAT) | 41.04 * | 16.07 * | 3.24 * |
| Biological index of fertility (BIF) | 87.41 * | 16.66 * | 22.0 * |

F test statistics and significance levels of two-way analysis of variance for the enzymatic activity associated with the cultivation system and terms of analyses fixed factors; * is $p = 0.05$, ns- not significant.

Stimulation of the development of microorganisms is accompanied by stimulation of the biochemical activity of the soil. The diversified methods of cover crop cultivation and wheat sowing caused noticeable changes in the activity of soil enzymes, which reflected environmental disturbances affecting both the soil and plants [31]. Dehydrogenase (DHA), catalase (CAT), acid (PAC) and alkaline (PAL) phosphatase are the soil enzymes whose activity are most often studied [32]. The dehydrogenase activity is considered an indirect indicator of the count and activity of soil microorganisms. As indicated by the mean dehydrogenase activity for two years (Figure 4), different cultivation methods used before sowing the cover crop and growing wheat significantly stimulated the activity of this enzyme. Increased enzymatic activity of dehydrogenase in the podzolic soil was observed in all treatments of the experiment, when compared with the conventional tillage (control) treatment. During the entire period of spring wheat growth, the highest dehydrogenase activity was observed in the fourth treatment, where the cover crop was sown directly and wheat was sown after ploughing tillage. The enzymatic activity was 76.9% greater than in the control treatment. There was also high dehydrogenase activity in the second treatment, where the cover crop was sown after skimming and then wheat was sown after tillage. There was another dependency concerning the dehydrogenase activity in the soil under the wheat plantation in unconventional cultivation treatments. The enzyme exhibited high activity during the third term of analyses when the seeds reached technological ripeness. The results from our study were in line with the findings reported in reference publications, which indicated higher dehydrogenase activity in the soil where simplified cultivation was applied [33]. Singh et al. [34] also indicated that a change in the soil microclimate caused by different residue of the leguminous cover crops and by the method of sowing crops influenced the metabolism of microorganisms, thus significantly contributing to the catabolic activity of soil dehydrogenase. According to Morris et al. [35], the mulch of stubble catch crops retains significant amounts of rainwater, and the soil moisture level affects the activity of this enzyme.

The applied methods of cover crop cultivation and spring wheat sowing also significantly influenced the acid phosphatase activity (Figure 4B,C). The enzyme exhibited lower activity in all treatments of unconventional cultivation. The lowest catabolic activity of acid phosphatase was observed in the second treatment, where the soil was skimmed for the cover crop and wheat was sown after tillage (Figure 4B). In comparison with the control treatment, acid phosphatase also exhibited significantly lower activity when the cover crop was sown directly and then wheat was sown after conventional tillage. Results also showed that the plants had the highest demand for phosphorus during the second term of analyses, i.e., at the flowering stage (BBCH 61–65). During this stage, it was observed that acid phosphatase activity was higher than during the other terms of analyses. The enzymatic activity was also high at the stage of full technological ripeness of the seeds in the third treatment, where the soil was skimmed before growing of the cover crop and then spring wheat was sown directly without tillage (Figure 4B).

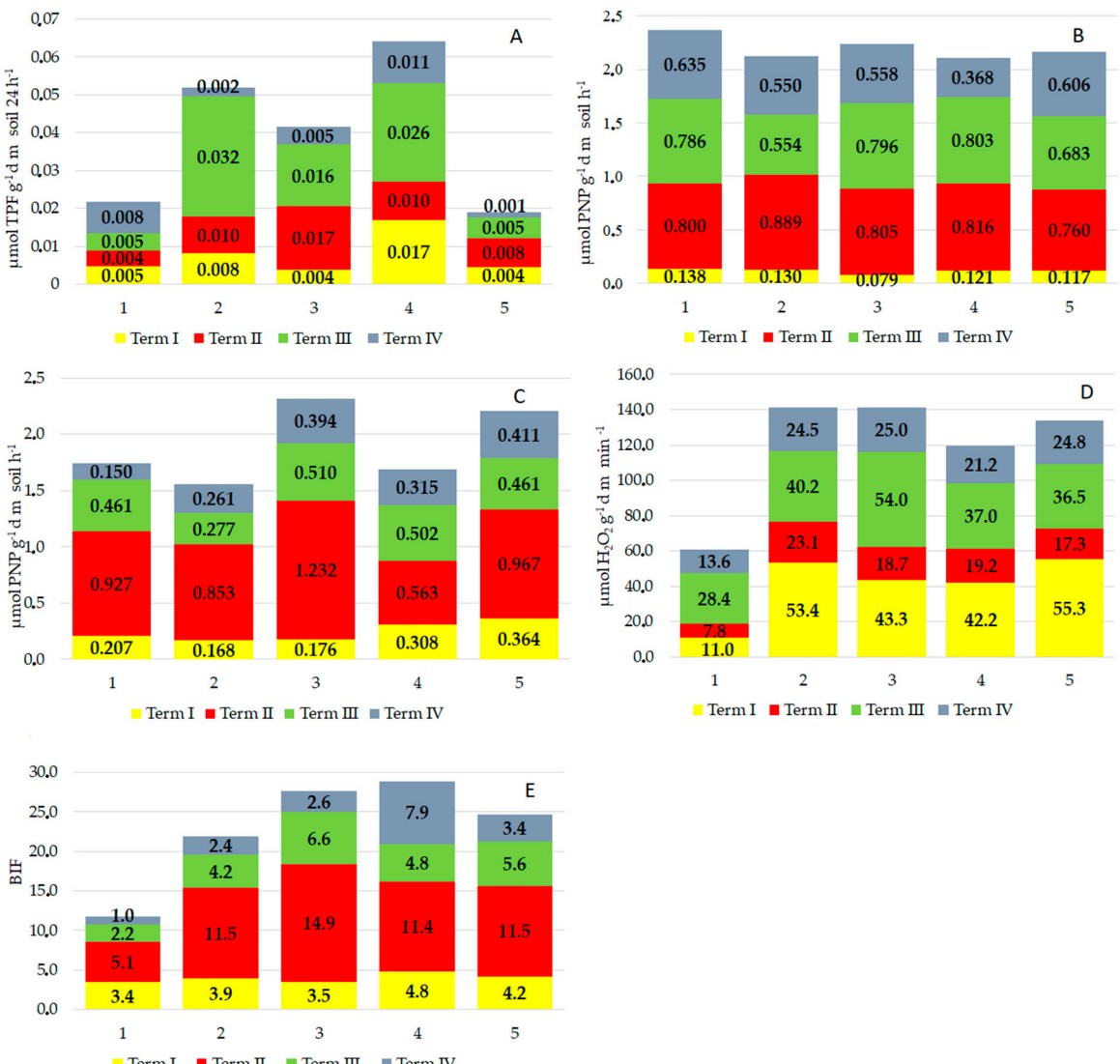

**Figure 4.** The enzymatic activity of the soil exposed to the pressure of the cultivation system under wheat. (**A**) dehydrogenase activity, (**B**) acid phosphatase activity, (**C**) alkaline phosphatase activity, (**D**) catalase activity and (**E**) biological index of fertility. 1—conventional tillage (CT) before sowing spring wheat without a cover crop, 2—conventional tillage (CT) skimming before sowing the cover crop and sowing spring wheat after conventional tillage (CT), 3—skimming before sowing the cover crop and sowing spring wheat directly (no-till (NT)) into mulch, 4—sowing the cover crop directly (NT) into the stubble and then sowing spring wheat after conventional tillage (CT) and 5—sowing the cover crop directly (NT) and sowing spring wheat directly into mulch (NT).

The alkaline phosphatase (PAL) activity also reacted significantly to the method of cover crop cultivation and wheat sowing (Figure 4C). There was a statistically significant increase in the enzyme activity in all the experimental treatments where unconventional cultivation was applied. The highest alkaline phosphatase activity was observed during the plants' flowering (2nd term of analyses). During the other terms of analyses, the catabolic activity of alkaline phosphatase was at a similar level.

The methods of cover crop cultivation and wheat sowing applied in the experiment had a positive effect on the acid phosphatase activity, which decreased (Figure 4B). When measuring the level of acid phosphatase activity in soil, it is necessary to remember that the roots of plants considerably influence the level of this enzyme in soil. They secrete significant amounts of acid phosphatase, especially when there is a phosphorus deficit [36]. The high concentration of acid phosphatase in the

control treatment, as compared with the other experimental treatments, may have been caused by the plants' response to phosphorus deficiency in the environment. Such dependencies were presented in the studies conducted by Lemanowicz and Koper [37] and Niewiadomska [38]. In another study, Lemanowicz and Koper [39] also observed higher catabolic activity of this enzyme in the experimental treatment without phosphorus fertilisation. Heflik et al. [40] found that the deficit of this macroelement stimulated plants' secretion of acid phosphatases.

The alkaline phosphatase activity increased significantly when different methods of cover crop cultivation and different methods of spring wheat sowing were applied (Figure 4C). This effect may have been caused by the activity of an increased count of soil microorganisms in order to leave an additional source of nutrients, including phosphorus, which came from the cover crops. Cover crop residue is not only a source of nutrients, but it also changes the physical properties of soil. The physical properties that may undergo changes are moisture, temperature and aeration. In consequence, it affects the microbiological and functional biodiversity of soil [41].

The catalase activity was significantly stimulated in all treatments of unconventional cover crop cultivation and wheat sowing, as compared with the control treatment (CT) (Figure 4D). During the entire growing season, the highest catalase activity was observed in the second treatment, where the soil had been skimmed before the cover crop was sown (CT skimming) and where spring wheat was sown after conventional tillage (CT). High catalase activity was also observed in the third treatment, where the soil had been skimmed before the cover crop was sown (CT skimming) and then wheat was sown directly (NT) into mulch. The catabolic activity of catalase in these treatments was at a similar level, i.e., 140 $\mu$mol $H_2O_2$ g$^{-1}$/dm/min$^{-1}$, and it was 57% higher than in the conventional tillage treatment. The high activity of this enzyme was observed during the first (emergence) and third (technological ripeness of wheat seeds) terms of analyses (Figure 4D).

The biological index of soil fertility (BIF), calculated on the basis of the dehydrogenase and catalase activity, was significantly higher in the unconventional cultivation treatments than in the conventional tillage treatments. It was the highest during the second term of analyses, when the plants were flowering. It ranged from 11.5 in the fifth and second treatments to 14.9 in the third treatment (Figure 4E).

The dependencies between the population of microorganisms and the enzymatic activity of soil after individual treatments and during individual terms of analyses were presented in the form of principal component analysis (PCA) (Figure 5). The first principal component explained 34% of variability, whereas the second explained 24% of the variability; i.e., 58% of the total variability. The PCA shows strong correlations between the soil biochemical parameters, such as DHA, PAL, PAC and BIF. It also indicates a strong positive correlation between the physiological groups of microorganisms (oligotrophs, copiotrophs, fungi and the total count of heterotrophic bacteria) analysed in the study. However, there was no strict dependence between the count of the microbial groups and the soil biochemical activity. The PCA shows a strong positive relation between the second term of analyses (BBCH 61–65) and the parameters of soil biochemical activity. These relations also apply to the third term of analyses (BBCH 87–89) but to a lesser extent. This can be explained by the fact that during the flowering period of wheat and during the milk maturity phase roots secrete amino acids, which are very valuable for the biological activity of soil. According to other publications, root secretions are an abundant source of carbon, nitrogen and energy, which significantly affect the biochemical activity of soil [42]. According to Pięta and Patkowska [43], wheat root secretions have the best quantitative and qualitative compositions of amino acids, because they contain large amounts of basic and aromatic amino acids. In addition, the substances secreted by wheat roots were found to directly affect the growth of the root systems. The PCA also indicated a negative relation between the amount of rainfall during the wheat-growing season and the count of *Actinobacteria*. Dry periods cause water deficits in soil, which increases the activity of *Actinobacteria*. There were similar negative relations between rainfall and the parameters of soil biochemical activity, DHA, PAL, PAC and BIF. However, there was no such relation for the catalase activity.

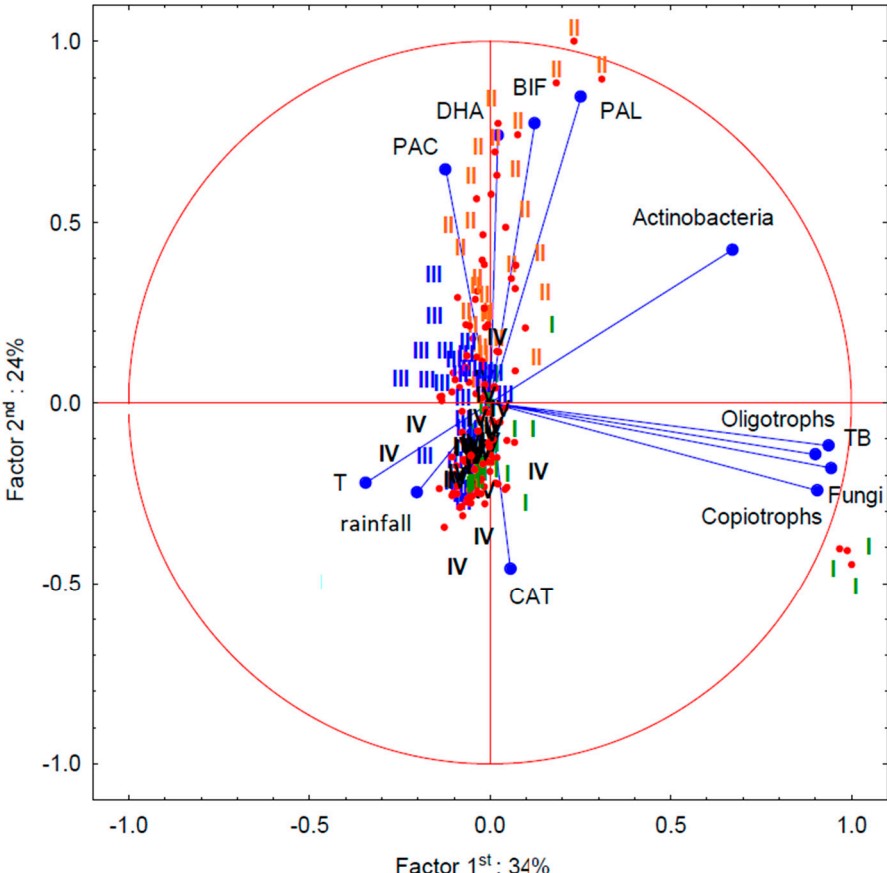

**Figure 5.** The relations between the soil microbiological activity after the applications of the cover crop and tillage methods. DHA—dehydrogenase activity, PAL—alkaline phosphatase activity, PAC—acid phosphatase activity, CAT—catalase activity, BIF—biological index of fertility, T—temperature, I—1st term (emergence) (BBCH 16–17), II—2nd term (flowering (BBCH 61–65) late June and early July), III—3rd term (technological ripeness of seeds) (BBCH 87–89) and IV—4th term (after the harvest).

Low-water content inhibits the growth of some groups of microorganisms. It reduces their populations and the populations of so-called inductive enzymes, which are secreted into soil by microorganisms. The results of studies conducted by Siwik–Ziomek and Szczepanek [44] and Niewiadomska et al. [38] confirmed the diversified dynamics of microbiological changes occurring during the growing season in different weather conditions. It is noteworthy that when the water level in soil is adequate, plants secrete various organic compounds, such as amino acids, carbohydrates and carboxylic acids. These substances are an initial, easily degradable source of carbon for microorganisms. They cause considerable proliferation of microorganisms, and thus, they increase the pool of soil enzymes. Secreted organic substances change the biophysical properties of soil, and thus, they increase the biochemical activity of microorganisms [45].

*3.6. Physiological State of Wheat*

The investigations of the carbon dioxide exchange revealed the highest net photosynthetic activity in the second treatment, where the cover crop was sown after skimming and where wheat was sown after conventional tillage. Similar results were observed in the third treatment, where the soil had been skimmed before the cover crop was sown and then wheat was sown directly (NT) into cover crop mulch (Figure 6).

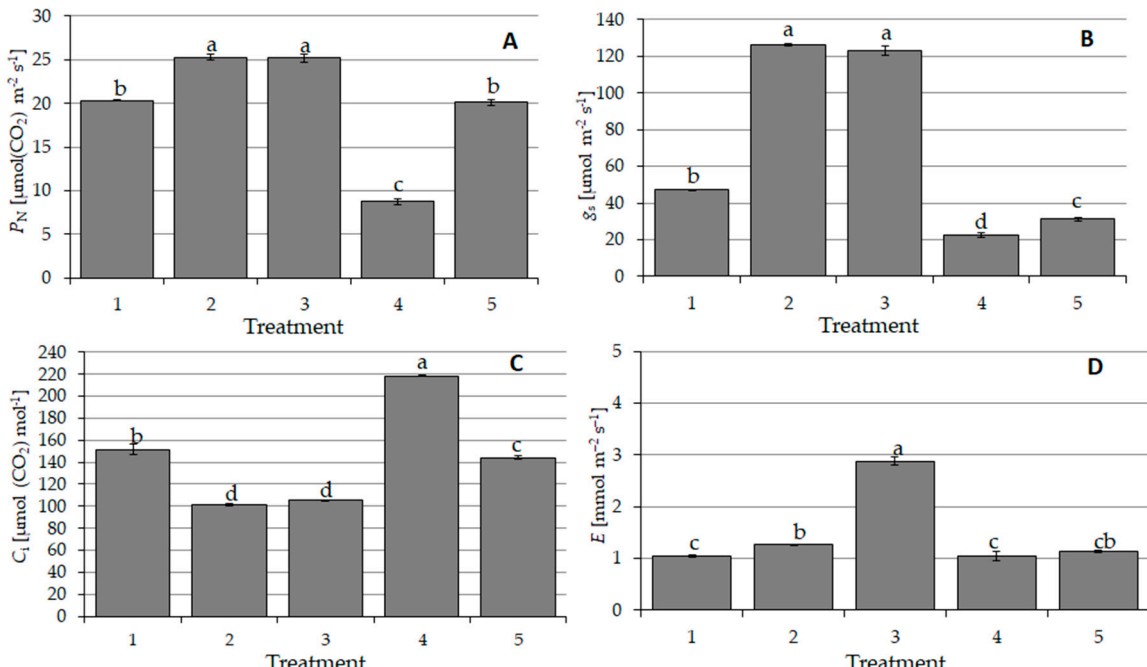

**Figure 6.** Gas exchange parameters (means ± SE): net photosynthesis rate—$P_N$ (**A**), stomatal conductance—$g_s$ (**B**), intercellular carbon dioxide ($CO_2$) concentration—$C_i$ (**C**) and transpiration rate—$E$ (**D**). a, b, c and d—different letters denote significant differences at level $\alpha$ = 0.05. 1—conventional tillage (CT) before sowing spring wheat without a cover crop, 2—conventional tillage (CT) skimming before sowing the cover crop and sowing spring wheat after conventional tillage (CT), 3—skimming before sowing the cover crop and sowing spring wheat directly (no-till (NT)) into mulch, 4—sowing the cover crop directly (NT) into the stubble and then sowing spring wheat after conventional tillage (CT) and 5—sowing the cover crop directly (NT) and sowing spring wheat directly into mulch (NT).

The lowest net photosynthetic activity was noted in the fourth treatment, where the cover crop was sown directly and wheat was sown after conventional tillage (Figure 6A). The same tendencies were observed for the stomatal conductivity, whereas the intercellular $CO_2$ concentrations exhibited the opposite tendencies (Figure 6B). Yang et al. [7] made similar observations in their study on the level of photosynthesis in simplified cultivation. As far as transpiration is concerned, there was a significantly higher level in the third treatment, where simplified tillage was applied; i.e., the soil was skimmed before sowing the cover crop and wheat was sown with no-till technology (Figure 6D).

The influence of the leguminous cover crops sown in various soil tillage systems and the influence of the wheat cultivation methods on the counts of selected groups of soil microorganisms, the biochemical activity of soil and the physiological state of the plants in correlation with the yield were visualised using a heat map (Figure 7). As the variability of the parameters exhibited similar trends in both years of the study (2014 and 2015), the mean values of these parameters in individual cultivation treatments were used in the heat map.

The comparison of the characteristics of all treatments showed that the first treatment (conventional tillage before wheat sowing—CT 1) and the second treatment (soil skimming before sowing the cover crop and conventional wheat sowing) were the most similar to each other. There were also strong similarities between the fourth and fifth treatments. In comparison with the other experimental treatments, the third treatment, where the soil had been skimmed before the cover crop was sown and wheat was sown directly (NT) into mulch, was characterised by much higher values of activity of the parameters under analysis.

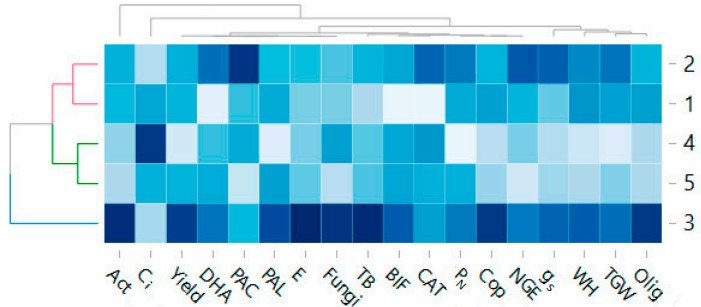

**Figure 7.** A comparable reaction between microbiological activity and gas exchange parameters and yield after using different soil cultivation systems. Abbreviation: Act—Actinobacteria, $C_i$—intercellular $CO_2$ concentration, Yield—t ha$^{-1}$, DHA—dehydrogenase activity, PAC—acid phosphatase activity, PAL—alkaline phosphatase activity, E—transpiration rate, TB—total count of heterotrophic bacteria, BIF—Biological index fertility, CAT—catalase activity, $P_N$—net photosynthesis rate, Cop—Copiotrophic bacteria, NGE—number of grains per ear, $g_s$—stomatal conductance, WH—weight of hectoliter, TGW—thousand-grain weight, Olig—Oligotrophic bacteria.

Results from the heat map showed that both the fourth and fifth treatments presented low values of the parameters characterising yield (NGE, TGW and WH) presented in another study (Figure 7). This effect may have been caused by the poor physiological state of the plants, as indicated by low values of the physiological parameters of wheat. Such parameters included $g_s$ (stomatal conductivity) and E (transpiration coefficient). In the fourth treatment, the net photosynthesis level ($P_N$) was also found to have low values. Reduced values of soil microbiological parameters (such as actinobacteria and oligotrophic bacteria) were used in these experimental treatments.

## 4. Conclusions

From the microbiological point of view, the research findings should be regarded as promising. Two years of field experiments with simplified or limited cultivation resulted in increased enzymatic, microbiological and physiological activity of the plants. The increase in the total count of heterotrophic and copiotrophic bacteria, as well as the increase in the dehydrogenase and catalase activity, which were observed after implementing cover crops and different tillage methods, indicated better soil fertility and productivity.

Additionally, the analysis of the 16S rRNA gene showed that the bacterial soil communities collected at term 'zero' and after the application of a particular cover crop cultivation method on a specific wheat sowing treatment differed in the structure and percentage distribution of individual taxa at the phylum level.

The simplification of the tillage system and the introduction of a cover crop increased the microbial biodiversity and soil moisture, as evidenced by a decrease in the count of actinobacteria in the soil. The occurrence of actinobacteria is characteristic for dry soils.

The best treatment for field production was the one in which the soil was skimmed before the cover plant was sown and wheat was sown directly into the mulch. The parameters measured from this treatment had the highest values when compared to all other treatments.

**Author Contributions:** Conceived and designed experiments—A.N., L.M., K.B., A.W.-M. and Z.W.; performed field experiments and analysed data—A.N., L.M., K.B., A.W.-M., Z.W. and R.G.; statistical analysis—A.B., Z.W. and K.B.; wrote the paper—A.N., L.M. and Z.W. and revised the manuscript—A.N., L.M., Z.W. and A.B.

**Funding:** This publication was co-financed within the framework of Ministry of Science and Higher Education programme as "Regional Initiative Excellence" in years 2019-2022, Project No. 005/RID/2018/19.

**Conflicts of Interest:** The authors declare no conflicts of interest.

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
