# Peer review of "The Influence of Tillage and Cover Cropping on Soil Microbial Parameters and Spring Wheat Physiology"

_agronomy, doi:10.3390/agronomy10020200_

Round 1

Reviewer 1 Report

Up to date study and interesting  results.

Several minor corrections  are necessary.

Line 57 - instead :" Due to the positive influence..."    better " Positive influence  of cover crop on the environment...

Line 74 - instead : Between 2014 and 2015  a static...."   should be "Field static experiment  was conducted  in 2014 and 2015."

Line 116 - instead ;"In both years... " should be just  "The average air temperature   ......... during the study".

Line 148 and 151,  please transfer the literature No  to the line 150 and 153, , respectively.

Line 157  the citation  of (15) should  be  as Stefanic et all.  (BIF ).

Line 342 instead of  "in 2007 Fierer ....."   simply   "Fierer et al (23)..."

Line 383 instead of "Also, Singh..."  better "Singh et al (29) also indicated..."

Line 453 instead "Both in 2014 and 2015..."  " Dehydrogenase (DHA)......

             .........both 2014 and 2015 ."

Author Response

All Reviewer suggestion was did.

Line 57 - instead :" Due to the positive influence..."    better " Positive influence  of cover crop on the environment...

Yes, it’s changed.

Line 74 - instead : Between 2014 and 2015  a static...."   should be "Field static experiment  was conducted  in 2014 and 2015."

Yes, it’s changed.

Line 116 - instead ;"In both years... " should be just  "The average air temperature   ......... during the study".

Yes, it’s changed.

Line 148 and 151,  please transfer the literature No  to the line 150 and 153, , respectively.

Yes, it’s changed.

Line 157  the citation  of (15) should  be  as Stefanic et all.  (BIF ).

Yes, it’s changed.

Line 342 instead of  "in 2007 Fierer ....."   simply   "Fierer et al (23)..."

Yes, it’s changed.

Line 383 instead of "Also, Singh..."  better "Singh et al (29) also indicated..."

Yes, it’s changed.

Line 453 instead "Both in 2014 and 2015..."  " Dehydrogenase (DHA)......

             .........both 2014 and 2015 ."

Yes, it’s changed.

Reviewer 2 Report

Global comments

The paper deals with the impacts of reduced tillage and incorporation of cover crop on soil microbial properties and crop productivity. Since these practices are assumed to provide benefits to agroecosystems (such as reduction of soil erosion, nutrient cycling…) and enhance sustainability of agriculture, a number of studies have already been done. In particular, these studies analyzed the effects of tillage systems on soil microbial diversity and activity. Therefore, even if the proposed work falls within the scope of the journal, it does not really provide new knowledge. As the study has been conducted over 2 years, it would have been more interesting to better analyze interactions between climatic conditions, tillage systems and cover crops on microbial variables. This would have given a more innovative character to the study. In fact, at field scale, the relative importance of climatic conditions vs agricultural practices as determinants of abundance, diversity and activities of soil microbial communities remains largely unresolved. This would have help to refine work hypotheses that are not clear for me.

Regardless of this, I suggest to authors to spend more time to clarify the design of their field experiment, as it is not clear.  More information on soil sampling also need to be provided. This could help to further analyze the results, by considering the evolution of plant rhizodeposition along wheat development. Finally, the authors do not address the potential impact of pesticides such as glyphosate and/or the quality of residues from cover crop (mix more or less deeply into the soil depending on soil tillage) on soil microbial communities and crop yield. Both the introduction and discussion could be improved to provide clarification on these aspects.

All these elements preclude the publication of this article in its present form.

Specific comments are also provided below.

Specific comments

Introduction

L67-70: Could you clarify why soil enzymes such as catalases, phosphatases and dehydrogenases have been measured? What were the assumptions? As cover crop were leguminous plants, it would have been interesting to measure soil enzymes implicated in nitrogen mineralization together with enzymes implicated in C mineralization. Please justify.

L68: “Mould” does not seem to be the right word. Please replace in all the document by fungi.

Material and Methods

L72-114. Could you described the experimental design (blocks? How many repetitions? Size of each plot?). What was the crop rotation?

In the experimental design, it is necessary to describe if and how wheat from cover crop treatments were fertilized. What could be the impact on soil microbial communities?

It is not clear for me if weed management was similar between the different treatments ? Is it the case?

L88. As this could have potential impacts on soil microbial communities and wheat yield, is the cover crop biomass and quality (C/N ratio…) differed between conventional tillage and non till systems?

L115. The paragraph considering weather data must be included in the results. However, would it have been possible to have average temperatures and rainfall measurements during wheat phenology? This would be better suited to explain variations in soil microbial densities, activities…

Table 1: rewrite “yellow”

L129-132. The authors have to describe in details how soils were sampled and they were process before microbial variables were measured?

L133-141. I don’t understand why the authors used different media to estimate density of total cultivable bacteria and copiotrophic / oligotrophic bacteria. It would have been better to use a medium such as TSA medium. Using this medium, it is possible to estimate total and copiotrophic versus oligotrophic bacteria, depending on the time of counting. Please justify.

Results and discussion

L226-228. I don’t understand how the authors can write this! The authors need to test if year has a significant influence on variables that have been measured or not. I didn’t see this in the results.

Table 3. The presentation of the results must be improved. What is the meaning of “terms of analysis”?

How the authors explain that for some treatments, the sum of the densities of copiotrophic and oligotrophic bacteria is higher than that of total heterotrophic bacteria?

I am surprized that the density of cultivable fungi does not change to much between till and no-till treatments. What are the assumptions? It would be interesting to have the 16S and 18S abundances and to analyze the variation of 16S/18S ratio in the different treatments.

L236-245. The authors observed that density of bacteria also varied depending of wheat phenology. What are the assumptions?

L318. Replace Firmecutes by Firmicutes in the text.

L322-328. The authors suggested that the higher relative abundance of Proteobacteria in no till systems could be due to the better water, air and nutrient conditions in soils compared to conventional systems. The authors should also make the link with the root development of wheat and the consequences on the entry of rhizodeposits into the soil that influence the availability of C.

L447-448. Considering the PCA analyses, I don’t understand why the authors didn’t test first if the two years can be discriminate. They could also include in their analyses climatic variables as illustrative ones. Please explain.    

The quality of Figure 5 must be greatly improved.

Physiological state of wheat. As the authors did analyses at the harvest of wheat, is it possible to include data of wheat yield?

Author Response

Dear Editor of Agronomy

Please find attached manuscript entitled „The Influence of the Soil Tillage Systems Applied Before Sowing a Stubble Cover Crop and Spring Wheat on the Microbiological Parameters of Soil and the Physiological State of Plants” corrected according reviewers suggestions:

Both the introduction and results discussion was improved according the Reviewer suggestions. All changes in the text are laballed at yellow and red color.

Specific comments:

(L115 was change:L231.) Table with climatic conditions was provided in results and discussion section for experiment period and with proper captions with indication on particular dates related to development stage of wheat.

In section results and discussion these data were related to results of biochemical soil activity parameters and microorganisms counts with the aid of PCA.

L: 248 Results and discussion section was expanded for results on organic carbon level and total nitrogen, as well as for C:N ratio.

L67-70: Could you clarify why soil enzymes such as catalases, phosphatases and dehydrogenases have been measured? What were the assumptions? As cover crop were leguminous plants, it would have been interesting to measure soil enzymes implicated in nitrogen mineralization together with enzymes implicated in C mineralization. Please justify

Catalases and dehydrogenases occur in the soil as an integral part of nienaruszonych life microorganisms cells and were presented in manuscript as measures of general microbial soil activity, as  well as were used to determine soil biological indicator of fertility (BIF) described by Steffanic. These are enzymes rom the class of oxidoreductases and play the most important functions in the environment. The second soil enzyme group are hydrolases, including phosphatases, which are included into phosphorus cycle. This was indicated in the manuscript according reviewer suggestion (L. 67-78  ).  Another enzymes included in this group are ureases, which indicate on intensity of nitrogen transformation in the soil and its availability to plants. Although reviewer suggested to include this group in the text, ureases were not planned to analyse in the presented investigations, hence it is not possible to present results in the manuscript. Anyway, we would like to thank for this valuable suggestion, which will be taken into consideration during planning the next experiment on effect of anthropopressure on soil environment.

L68: “Mould” does not seem to be the right word. Please replace in all the document by fungi.

In all document word  “mould” was replace to ‘fungi”

L72-114 The experimental design was describe ((Blocks, How many repetitions? ,Size of each plot? and What was the crop rotation? Now L 95-98. 

In the manuscript  nutrition for wheat was only indicated (L.125-126). There was no fertilizing under intercrop. It was also indicated in section of Materials and methods that all chemical treatments (plant nutrition and pesticides) were performed according good agricultural practice for spring wheat and were not here the influencing factor.

In response to reviewer comment we would like inform that weed management was not similar between different treatments, including Glyphosate doses which are labelled in table 2 (L 103). The other herbicide doses were similar.

L129-132. The authors have to describe in details how soils were sampled and they were process before microbial variables were measured?

Corrected in the text according Reviewer suggestion.

The method of collecting soil samples and preparing them for biochemical microbiological analyzes has been supplemented in the paper L.140 - 143

L133-141 I don’t understand why the authors used different media to estimate density of total cultivable bacteria and copiotrophic / oligotrophic bacteria. It would have been better to use a medium such as TSA medium. Using this medium, it is possible to estimate total and copiotrophic versus oligotrophic bacteria, depending on the time of counting. Please justify.

Such media, which we used to determine oligotrophs (NB) and copiotrophs (DNB), are widely used in the experiment and presented in the literature. We use the TSA medium mainly for the determination of microorganisms isolated from water or animal feed and for Enterobacteriaceae. However, the Reviewer's suggestion and methodology is very interesting and we will definitely learn about it in the future and use it in other experiments. L.151-152. 

L226-228. I don’t understand how the authors can write this! The authors need to test if year has a significant influence on variables that have been measured or not. I didn’t see this in the results.

No significant differences were found for the parameters studied between years, therefore, years were treated as replicates. Furthermore, the results were analysed with two-way ANOVA using Statistica 9.1 software, where the cultivation system and the term of analysis were the factors differentiating the traits under study to estimate the soil biochemical activity parameters. This was explained and indicated in the section on statistical analysis.

The order of table was change because new tables were added. Table 3. The presentation of the results must be improved. What is the meaning of “terms of analysis”? now is Table 5 . The table was improved. The terms of analysis were labelled in the table.

How the authors explain that for some treatments, the sum of the densities of copiotrophic and oligotrophic bacteria is higher than that of total heterotrophic bacteria?

 It is well known that media with a low concentration of organic substances (media with a low nutrient content) give a larger number of colonies than media with a high concentration of organic substances (media with a high content of nutrients). Studies show that nutrients with a high content of nutrients not only allow the rapid growth of some organisms, but also inhibit the growth of many soil organisms. Nutrient media with high nutrient content are used to study the physiological and taxonomic properties of isolates from water and land sources as well as animal bodies. Broth medium (NB) is often used for many types of bacteria, which is also commonly used to count and test soil bacteria such as copyotrophs. HATTORI observed that more colonies were obtained by diluting NB medium with soil samples, and found many bacteria sensitive to NB medium on diluted NB plates.

However, standard agar medium used by us to estimated heterotrophic bacteria contains an optimal level of nutrients between NB and DNB medium and hence sum densities of copiotrophic and oligotrophic bacteria was higher than that of total heterotrophic bacteria.

I am surprized that the density of cultivable fungi does not change to much between till and no-till treatments. What are the assumptions? It would be interesting to have the 16S and 18S abundances and to analyze the variation of 16S/18S ratio in the different treatments.

Based on Martin medium measurement we could estimate only saprotroph. Previous research indicated that the use cover cropping caused an increase of species diversity, while the use of no-till caused shift the symbiotroph to saprotroph ratio to favor symbiotrophs. These management-induced shifts in fungal community composition could lead to greater ecosystem resilience and provide greater access of crops to limiting resources. For estimation of symbiotrophs, where mycorrhizal fungi and fungi from Trichoderma genus the nutrient reach media are used. It was indicated in the text L.:344-361. Suggestion of reviewer concerning analysis 18S is correct, however we did not do it in our investigations. However, the direction of changes in the soil in case of analysed fungi with the aid of Martin medium, as a result of covercrop, as well as type of cultivation of covercrop and wheat cultivation were revealed.

L236-245. The authors observed that density of bacteria also varied depending of wheat phenology. What are the assumptions?

According to state of the art. the roots secretions are perfect source of carbon, nitrogen and energy. According Pięta and Patkowska roots secretion of wheat are characterised by the most appropriate quantitative and qualitative of aminoacids, because they include great amounts of alkaline and aromatic aminoacids. Moreover, the release of substances from wheat roots was found, which was directly related to the growth of the root system. This was also taken into consideration the manuscript. It was indicated in L. 562-566

The PCA was changed and explained according to the reviewer's suggestions

The crop results were not presented in this manuscript, because they will be part of another manuscript. In this manuscript yield of wheat were only used to show correlations.

This article is original and is not being considered for publication elsewhere.

This work language was corrected by native speaker from United Kingdom, Robert Kippen. Me and co-authors hope, this work will be interested for You and potential Reviewers.

I have prepared this manuscript according to suggestion, but if you have any questions or suggestions regarding our paper, do not hesitate to contact me. I will do my best to correct this work and send for you most quickly.

Best Regards

Alicja Niewiadomska
